# Differential Mitochondrial, Oxidative Stress and Inflammatory Responses to SARS-CoV-2 Spike Protein Receptor Binding Domain in Human Lung Microvascular, Coronary Artery Endothelial and Bronchial Epithelial Cells

**DOI:** 10.3390/ijms25063188

**Published:** 2024-03-10

**Authors:** Gabrielė Kulkovienė, Deimantė Narauskaitė, Agilė Tunaitytė, Augusta Volkevičiūtė, Zbigniev Balion, Olena Kutakh, Dovydas Gečys, Milda Kairytė, Martyna Uldukytė, Edgaras Stankevičius, Aistė Jekabsone

**Affiliations:** 1Department of Drug Chemistry, Pharmacy Faculty, Lithuanian University of Health Sciences, 50162 Kaunas, Lithuania; gabriele.kulkoviene@lsmu.lt; 2Preclinical Research Laboratory for Medicinal Products, Institute of Cardiology, Lithuanian University of Health Sciences, 50162 Kaunas, Lithuania; deimante.narauskaite@lsmu.lt (D.N.); agile.tunaityte@lsmu.lt (A.T.); augusta.volkeviciute@lsmu.lt (A.V.); zbigniev.balion@lsmu.lt (Z.B.); olena.kutakh@lsmu.lt (O.K.); milda.kairyte@stud.lsmu.lt (M.K.); martyna.uldukyte@stud.lsmu.lt (M.U.); edgaras.stankevicius@lsmu.lt (E.S.); 3Institute of Pharmaceutical Technologies, Faculty of Pharmacy, Lithuanian University of Health Sciences, 50162 Kaunas, Lithuania; dovydas.gecys@lsmu.lt; 4Laboratory of Molecular Cardiology, Institute of Cardiology, Lithuanian University of Health Sciences, 50162 Kaunas, Lithuania

**Keywords:** COVID-19, mitochondrial network, cristae, STED microscopy, mitochondrial superoxide, ROS, inflammation

## Abstract

Recent evidence indicates that the SARS-CoV-2 spike protein affects mitochondria with a cell type-dependent outcome. We elucidate the effect of the SARS-CoV-2 receptor binding domain (RBD) on the mitochondrial network and cristae morphology, oxygen consumption, mitoROS production, and inflammatory cytokine expression in cultured human lung microvascular (HLMVECs), coronary artery endothelial (HCAECs), and bronchial epithelial cells (HBECs). Live Mito Orange staining, STED microscopy, and Fiji MiNa analysis were used for mitochondrial cristae and network morphometry; an Agilent XFp analyser for mitochondrial/glycolytic activity; MitoSOX fluorescence for mitochondrial ROS; and qRT-PCR plus Luminex for cytokines. HLMVEC exposure to SARS-CoV-2 RBD resulted in the fragmentation of the mitochondrial network, mitochondrial swelling, increased cristae area, reduced cristae density, and suppressed mitochondrial oxygen consumption and glycolysis. No significant mitochondrial morphology or oxygen consumption changes were observed in HCAECs and HBECs. SARS-CoV-2 RBD induced mitoROS-mediated expression of cytokines GM-CSF and IL-1β in all three investigated cell types, along with IL-8 expression in both endothelial cell types. The findings suggest mitochondrial ROS control SARS-CoV-2 RBD-induced inflammation in HLMVECs, HCAECs, and HBECs, with the mitochondria of HLMVECs being more sensitive to SARS-CoV-2 RBD.

## 1. Introduction

Many COVID-19 patients suffer from upper respiratory tract disease or pneumonic complications, where severe cases can lead to respiratory failure, shock, and even death [1]. While respiratory symptoms are the most common among patients, there is evidence that COVID-19 can also impact the cardiovascular system, causing serious complications such as myocardial injury, myocarditis, and endothelial dysfunction [2]. Some studies have suggested that COVID-19 causes severe pulmonary and cardiovascular complications via mitochondrial signalling and immuno-metabolic reprogramming-mediated inflammation [3].

The virus enters cells through the angiotensin-converting enzyme 2 (ACE2) receptor, with TMPRSS2 facilitating entry by cleaving the spike glycoprotein. ACE2 is present in various tissues, including the lung, heart, vessels, and liver and binds to the virus’s receptor binding domain (RBD), leading to endocytosis. Once inside the cell, viral RNA and proteins localise to the mitochondria, which are closely connected to pathways regulating inflammation [4,5,6]. SARS-CoV-2 infection disrupts the mitochondrial network, alters respiratory chain function, and increases mitochondrial reactive oxygen species (mitoROS) [7,8,9], leading to oxidative stress in severe cases of COVID-19 [10]. It has been reported that SARS-CoV-2 can trigger a feedback loop that causes persistent inflammation and cellular damage via mitochondrial dysfunction and oxidative stress [11]. This can affect cells that express high levels of ACE2, such as endothelial and epithelial cells.

Despite the previously observed effects associated with mitochondrial involvement in inflammation, the exact mechanism of mitochondria-dependent signalling pathways in acute inflammation during COVID-19 remains unknown and requires further exploration. This study aims to investigate and compare the involvement of mitochondria and mitoROS in the inflammation triggered by the SARS-CoV-2 spike glycoprotein receptor binding domain (SCoV2-RBD) in human lung microvascular and coronary artery endothelial and bronchial epithelial cells, with a specific focus on the mitochondrial network integrity and cristae density.

## 2. Results

### 2.1. The Effect of SCoV2-RBD on Mitochondrial Morphology

First, we evaluated changes in the mitochondrial morphology of human lung microvascular endothelial (HLMVECs), coronary artery endothelial (HCAECs), and bronchial epithelial cells (HBECs) after 24-h SCoV2-RBD treatment. Visual examination of the mitochondrial network revealed higher fragmentation in SCoV2-RBD-affected HLMVECs (Figure 1a). Treatment with SCoV2-RBD decreased the mitochondrial area, or footprint, by 25%, the length of the branches by 11.6%, summed branch length by 20.9%, and the number of network branches by 25.9% compared to healthy cells (Figure 1b). No visual qualitative or significant quantitative changes were observed in HCAECs and HBECs after SCoV2-RBD treatment (Figure A4 and Figure A5).

Moreover, the intercristae distance in SCoV2-RBD-primed HLMVECs increased significantly by 30% compared to the untreated cells (Figure 2). Additionally, they exhibited an increase in the average crista area by 20% and mitochondrial thickness by 18% compared to the control area. However, no structural alterations in the mitochondrial cristae were observed in HCAECs and HBECs.

### 2.2. The Effect of SCoV2-RBD on Mitochondrial and Glycolytic Activity and mitoROS

Next, the study evaluated mitochondrial respiration and glycolysis activity after priming HLMVEC, HCAEC, and HBEC cultures with SCoV2-RBD. The mito-morphological events in HLMVECs were accompanied by the significant suppression of mitochondrial respiration and glycolysis (Figure 3). A significant decrease was observed in basal, maximal, ATP-linked oxygen consumption, and glycolysis. However, any substantial changes in the energetic activity of HCAECs and HBECs were determined. These results suggest that SCoV2-RBD impairs mitochondrial function in HLMVECs by disrupting their network and cristae morphometrics.

Changes in mitochondrial respiratory chain activity might affect ROS production. Therefore, further in this study, we have investigated the effect of the SCoV2-RBD protein on mitoROS generation in HLMVECs, HCAECs, and HBECs. The primary ROS produced within mitochondria is a superoxide, which cannot cross membranes and stays at the production site. We assessed mitochondrial superoxide with a fluorescent probe MitoSOX red. SCoV2-RBD significantly induced mitoROS production in HLMVECs and HCAECs: MitoSOX fluorescence intensity increased by 2.6% and 7.4%, respectively, compared to the untreated cells (Figure 4). No change in mitoROS generation in HBECs was detected.

### 2.3. The Impact of mitoROS on SCoV2-RBD-Induced Effect on Mitochondrial Morphology

mitoROS production might result from changes to the mitochondrial network morphology, but it could also be the cause of such changes. We applied the mitochondrial superoxide scavenger MitoTempo to test whether mitoROS influenced the mitochondrial network changes after SCoV2-RBD treatment in HLMVECs. Interestingly, MitoTempo rescued mitochondrial fragmentation in HLMVECs (Figure 5a) and restored the summed branch length without affecting the mean branch length (Figure A6). The latter parameter indicates the overall length of branches, while the summed branch length represents a sum of all branch lengths divided by the number of individual structures. This finding suggests a decrease in the number of individual networks, indicating restoration of the interconnectivity of the network. Such an indication is supported by the finding that MitoTempo prevents the loss of network branches. However, mitoROS had no detectable impact on the SCoV2-RBD effect on the mitochondrial footprint area (Figure A6), suggesting no control for mitoROS in SCoV2-RBD-induced mitochondrial loss.

### 2.4. The Impact of mitoROS on SCoV2-RBD-Induced Inflammatory Cytokines

mitoROS are involved in multiple cell signalling pathways, including inflammation. In the study, we further tested whether inflammatory cytokine expression and secretion after SCoV2-RBD treatment was affected by the mitoROS scavenger MitoTempo in HLMVECs, HCAECs, and HBECs.

SCoV2-RBD treatment resulted in significant cytokine upregulation in all cell lines compared to the untreated cells (Figure 6). In HLMVECs, NF-kβ expression increased by 1.15-fold, GM-CSF even by 4.25, IL-8 by 1.5, and IL-1β by 3.1. In HCAECs, GM-CSF expression increased by 3.5-fold, IL-8 by 1.35, IL-1β by 1.9, and IL-6 by 1.3. In HBECs, the average NF-kβ expression level increased by 1.2-fold, CCL2 by 2.3, GM-CSF by 5.1, IL-8 by 1.7, and IL-1β by 4.6.

The addition of MitoTempo during SCoV2-RBD treatment significantly prevented SCoV2-RBD-induced gene upregulation down to the baseline value or even induced downregulation. Significant differences between SCoV2-RBD and MitoTempo-supplemented SCoV2-RBD treatment were observed in GM-CSF (fold changes or FC 4.25 vs. 2.5) and IL-8 (FC 1.5 vs. 0.6) in HLMVECs; GM-CSF (FC 3.5 vs. 0.9), IL-8 (FC 1.35 vs. 1.2), and IL-1β (FC 1.9 vs. 1.0) in HCAECs; GM-CSF (FC 5.1 vs. 1.4) and IL-1β (FC 4.6 vs. 1.0) expression in HBEC cells. Of note, the addition of MitoTempo significantly downregulated the expression of IL-8 (FC 0.6) in HLMVECs and NF-kβ (FC 0.8) in HCAECs. Despite the attenuating effect of MitoTempo on cytokine expression, GM-CSF and IL-1β remained significantly upregulated in HLMVECs. Similar results were observed in IL-8 and IL-6 expression in HCAECs and IL-8 and NF-kβ expression in HBECs.

SCoV2-RBD treatment significantly increased GM-CSF secretion by HLMVECs and HCAECs to their culture medium in a MitoTempo-sensitive manner, clearly indicating the involvement of mitoROS in the GM-CSF pathway (Figure 7). In contrast, SCoV2-RBD did not change the secretion of other investigated cytokines and even decreased the secreted level of TNF-α in HLMVECs (Figure A7). Furthermore, the levels of IL-1β, IL-6, and TNF-α were below the detection limits in both the control and SCoV2-RBD-treated samples of HBECs (Figure A7).

## 3. Discussion

### 3.1. HLMVECs Respond to SCoV2-RBD by Rapid Mitochondrial Changes

The study revealed the exclusive sensitivity of HLMVEC mitochondria to SCoV2-RBD signalling. Treatment with the SCoV2-RBD protein resulted in a fragmented mitochondrial network (shorter, less branched structures) and a decreased mitochondria-covered area, indicating a loss of total mitochondrial volume. Moreover, the remaining mitochondria after SCoV2-RBD treatment exhibited swelling, as evidenced by thicker mitochondrial filaments, which correlated with increased intercristae spacing and cristae area. Mitochondrial respiratory efficiency relies on assembling and maintaining mitochondrial respiratory supercomplexes, stabilising the individual complex structure and their performance by substrate channelling [8]. The best-known mitochondrial supercomplexes are respirasomes composed of complexes I, III, and IV [9]. The mitochondrial capacity to keep supercomplexes assembled depends directly on the cristae shape, namely, the tightness of cristae invaginations and the distance between the cristae-forming membranes [10]. Thus, an increase in the average crista area after HLMVEC interaction with SCoV2-RBD suggests that mitochondrial respiratory chain complex efficiency might be suppressed by destabilising them due to the loss of cristae compactness. Indeed, this structural alteration was accompanied by suppressed basal, maximal, and ATP-linked mitochondrial respiration, pointing to inhibition of the mitochondrial respiratory chain complexes or respiratory chain-supporting processes such as the Krebs cycle. Considering that both coupled and uncoupled mitochondrial respiration was decreased to the same extent, it is unlikely that SCoV2-RBD signalling affected the phosphorylation machinery of the cells. Similar mitochondrial impairment and fragmentation by SARS-CoV-2 subunit 1 (which also contains the receptor binding domain) was recently observed in primary human ventricular cardiomyocytes [12]. Additionally, mitochondrial respiratory chain dysfunction was followed by Drp1-mediated fission and AIF-induced apoptosis in Calu-3 (human lung adenocarcinoma-derived airway epithelial) cells after SARS-CoV-2 infection [13]. However, no other studies have been published evaluating the effect of the SARS-CoV-2 virus or its components on mitochondrial cristae area and density.

### 3.2. Mitochondrial Signaling in SARS-CoV-2 Infection Is Cell and Pathway-Specific

Investigating mitochondrial responses to SCoV2-RBD treatment unveiled intriguing cell-type specificity, challenging existing perceptions of consistent mitochondrial alterations across cell lines. Despite HLMVEC mitochondria showing dramatic changes over 24 h with SCoV2-RBD, no network, cristae, or activity changes were detected in HCAECs and HBECs after the same treatment. Although this might seem controversial in the context of numerous reports regarding SARS-CoV-2-induced mitochondrial damage in vascular endothelial and airway epithelial cells, the explanation might be found in the differences between the experimental models, including cell types, infection-mimicking approaches (from full virus to spike protein, its subunits, domains or epitopes) as well as treatment duration and concentration. When the spike protein and its domains mainly refer to mitochondrial signalling via ACE2 [8,12], viral infection activates the mitochondrial immune response via recruitment of the mitochondrial antiviral signalling protein MAVS, leading to the inhibition of mitochondrial fission and mitochondrial network elongation [14]. Nevertheless, our study indicates that certain cell types, including lung capillary endothelial cells, are more sensitive to SCoV2-RBD signalling to their mitochondria, and this sensitivity might contribute to SARS-CoV-2 (and, potentially, other similar viruses)-induced alveolar damage leading to hypoxemia.

### 3.3. SARS-CoV-2-RBD Suppresses Energetic Metabolism in HLMVECs

Our investigation contributes additional knowledge regarding the energetic metabolism changes in lung capillary endothelial cells during SARS-CoV-2 infection. While SARS-CoV-2 was previously described to induce a mitochondrial to glycolytic metabolic shift in these cells [8,15], here, we observed a slow-down in both glycolysis and mitochondrial respiration in HLMVECs, indicating that the cells are switched to a low-energy profile. Cell viability (tested by PrestoBlue metabolic activity assay) of all three cell types after 24-h treatment with SCoV2-RBD was unchanged; thus, cell loss did not influence the results. Although COVID-19 infection is reported to increase glycolysis via metabolic changes similar to the Warburg effect in cancerous cells [16], ACE2 signalling has the opposite effect, leading to the downregulation of glycolytic enzymes and glycolysis suppression [17], suggesting the same or a similar mechanism taking place in HLMVECs under SCoV2-RBD treatment. However, glycolytic efficiency was unchanged in HCAECs and HBECs after 24-h treatment with SCoV2-RBD, indicating that such glycolysis suppression is not the case in bronchial epithelial and coronary endothelial cells.

### 3.4. mitoROS Mediates SARS-CoV-2-RBD-Induced Mitochondrial Fragmentation in HLMVECs

Our study revealed differential mitoROS production patterns across different cell types, offering insights into the divergent susceptibility of cardiovascular and pulmonary tissues to SARS-CoV-2 infection. Lower cristae density might lead to increased cristae volume, which, in turn, directly correlates to mitoROS production intensity [18]. Indeed, in HLMVECs, SCoV2-RBD-induced mitochondrial suppression was accompanied by an increase in mitoROS. Moreover, mitoROS partially mediated the mitochondrial network changes after SCoV2-RBD treatment; they appeared to cause fragmentation (summed branch length and branch number was MitoTempo-sensitive, Figure 5) but not mitochondrial degradation (no MitoTempo effect on the mitochondrial footprint and mean branch length, Figure A6). Interestingly, despite no detectable mitochondrial functional or structural changes, an even more prominent mitoROS increase was induced in HCAECs. There was no mitoROS increase in HBECs after 24 h with SCoV2-RBD; however, this does not exclude the possibility that mitoROS were generated more intensively during some treatment periods before returning to the control level. ACE2 binding can trigger mitoROS via NOX4 [19]; thus, the observed variation pattern in mitoROS production induced by SCoV2-RBD—high degree in HCAECs, lower degree in HLMVECs, and no change in HBECs—could be attributed, in part, to the differential expression of ACE2 in cardiovascular tissue compared to pulmonary and in endothelium compared to epithelium [20,21].

### 3.5. mitoROS Mediates Cytokine Production in Response to SARS-CoV-2-RBD

This study provided evidence about the pivotal role of mitoROS signalling in regulating key cytokine expression patterns during SARS-CoV-2 infection. Notably, the mitoROS scavenger MitoTempo prevented SCoV2-RBD-induced expression of GM-CSF and IL-1β in all three investigated cell types and IL-8 in HLMVECs and HCAECs, indicating mitoROS signalling as a key regulator of the genes after RBD–receptor interaction. Moreover, mito-ROS-dependent induction of GM-CSF expression was confirmed by the secretion of this cytokine in HCAECs. In normal conditions, GM-CSF regulates alveolar clearance; however, during severe infections, it recruits inflammatory cytokines [22]. Overproduction of GM-CSF is related to harmful hyperinflammatory responses to COVID-19 [23,24] and can lead to endothelial dysfunction [25]. However, the changes in GM-CSF secretion were not observed in HLMVEC and HBEC cell cultures. Moreover, there were no changes in the level of other investigated cytokines after SCoV2-RBD treatment, and in HBEC cultures, the level of the secreted cytokines was extremely low. The fact we did not observe a concentration change after 24 h of treatment with SCoV2-RBD might be due to an assessment timing mismatch with the cytokine secretion peaks. Additionally, cytokine synthesis and secretion might be controlled by different mechanisms, the expression being related to preparation for the immune response and the secretion participating in the immune response after a particular trigger.

In summary, the study revealed that SCoV-2-RBD triggers inflammatory gene upregulation in endothelial and epithelial cells primarily through the mitoROS – GM-CSF pathway, which in HLMVECs are also accompanied by acute mitochondrial network, cristae structure, and function changes. The findings suggest mitoROS scavengers and/or antioxidants are potential therapeutics for preventing severe complications during coronavirus infections.

## 4. Materials and Methods

### 4.1. Cell Culture and Treatments

Primary human lung microvascular and coronary artery endothelial cells (HLMVECs and HCAECs, Cell Applications, San Diego, CA, USA), and hTERT-immortalized bronchial epithelial cells (HBECs, Evercyte, Wien, Austria) were grown in Microvascular Endothelial Cell Growth Medium (Cell Applications), MesoEndo Cell Growth Medium (Cell Applications), and MEMα (Gibco, Life Technologies, Thermo Fisher Scientific, Carlsbad, CA, USA), respectively, supplemented with 10% FBS and 1% penicillin/streptomycin. MEMα for HBECs also contained 5 ng/mL EGF and 1 µg/mL hydrocortisone. Cells were cultured until 70–80% confluency, plated for experiments, and 24 h later treated with the SCoV2-RBD, 2.8 μg/mL; Baltymas, Vilnius, Lithuania) with and without a specific mitochondrial superoxide scavenger MitoTempo (10 μM; Sigma-Aldrich, Saint Louis, MO, USA) for 24 h.

### 4.2. Mitochondrial Morphology

A total of 6 × 10^4^ cells were seeded onto the glass part of P35 confocal dishes (75856-742, VWR), treated, stained with 100 nM Mito Live Orange (Abberior, Goettingen, Germany) for 40 min, and imaged using a Zeiss Axio Observer.Z1 fluorescent microscope. Mitochondrial morphology was analysed using the Fiji MiNa toolset (Figure A1). Images were preprocessed using unsharp masking, CLAHE, and median filtering. The MiNa plugin was used to calculate the area of mitochondrial structures and produce a morphological skeleton for calculating the following parameters: mean branch length, summed branch length mean, and mean of network branches [26]. Cristae were observed by an Olympus IX83 microscope (Olympus Corporation, Tokyo, Japan) with a STEDYCON STED nanoscope (Abberior, Goettingen, Germany) and UPLXAPO100xO NA1.45 objective. Parameters: pixel sizes 10–20, 2–5 line steps, dwell times 10 µs, pinhole 40 µm. Images were deconvoluted using Huygens software. The spacing between cristae was estimated by measuring the distances between fluorescence peaks along the 1–2 µm length intensity line profiles. Intensity profiles were plotted across the mitochondrion width to assess the thickness of mitochondrial filaments. Three line profiles were measured for intercristae distance evaluation, while another three line profiles were utilised to assess the mitochondrial thickness in each cell image. For the average crista area measurements, the cristae were segmented using the Trainable Weka Segmentation plugin in Fiji, following the guidelines outlined by Segawa et al. [27]. The plugin generated a probability map (Figure A2) highlighting areas likely to represent cristae against the background. The maps were then subjected to thresholding, manually adjusted to exclude non-focused mitochondrial filaments, and processed using the “Watershed” function to separate connected components, resulting in a binary mask. The “Analyze particles” function was employed to outline the regions of interest. Subsequently, these structures were superimposed onto the original image for measurement, and the average area of all outlined cristae was calculated. The mitochondrial network parameters were assessed in at least 11 cells per group per three experiments, evaluating the entire mitochondrial network within each cell. For cristae morphometrics data, 14–16 cells were examined for each group in three experimental repeats, with analysis carried out on randomly selected regions.

### 4.3. Mitochondrial and Glycolytic Activity

Cells were seeded into the Seahorse XFp well plates at a density of 6 × 10^3^ cells/well. Mitochondrial and glycolytic function was assessed by the Seahorse XFp analyser, Cell Mito Stress Test Kit (Agilent Technologies, Santa Clara, CA, USA). Final inhibitor concentrations in the wells were 1.5 μM oligomycin, 0.5 μM FCCP, 0.5 μM antimycin A, 0.5 μM rotenone. The oxygen consumption rate (OCR) and extracellular acidification rate (ECAR) were normalised to total protein content determined by the Bradford assay. Optical density after reaction with the Bradford reagent (Merck, Rahway, NJ, USA) was assessed by a Varioskan™ LUX Multimode Microplate Reader (Thermo Fisher Scientific Inc., Waltham, MA, USA). The data were analysed by Wave vs. 2.6.1. software.

### 4.4. Mitochondrial Superoxide

Cells were seeded in 96-well plates at 104 cells/well. After the treatments, the cells were stained with MitoSox Red (Thermo Fisher Scientific) for 30 min in HBSS at 37 °C and imaged with an Olympus fluorescent microscope APX100. The mitochondrial location of MitoSOX Red was confirmed by co-staining with 500 nM MitoTracker Green (Thermo Fisher Scientific, Figure A3). Although MitoSOX detection by fluorescence has some limitations due to the lack of a fully specific adduct formation [28], control measures were implemented, and the minimum MitoSOX concentration was used, as suggested by Murphy and colleagues [29]. Antimycin A (50 μM) was used 30 min prior to cell staining as a positive control and MitoTempo [30] (10 μM) as a negative control. The fluorescence intensity of the cells was measured using Olympus cellSens software, selecting cell area segmentation from the background function (Figure 2b). Each measurement was performed in triplicate, with at least 10 images taken per replicate.

### 4.5. Gene Expression

Cells were seeded in 12-well plates at 10^5^ cells/well, treated, total RNA extracted from the cells using PureLink RNA Mini Kit and reverse-transcribed by the High-Capacity cDNA Reverse Transcription Kit. Real-time quantitative PCR with Power SYBR Green Chemistry was used to evaluate the expression of IL-6, IL-8, IL-1β, NF-kβ, GM-CSF, and CCL2 genes. The GAPDH gene was used as the endogenous control. Sequences of the PCR primers used in gene expression evaluation are presented in Table A1. Gene expression changes were normalised to untreated cells and represented as fold changes. All reagents were from Thermo Fisher Scientific.

### 4.6. Cytokine Secretion Analysis

GM-CSF, IL-1β, IL-6, CCL2, and TNF-α levels in the supernatants were measured using a multiplex human Procarta Plex 5-plex Kit (Thermo Fisher Scientific), according to the manufacturer’s instructions. Briefly, after treatment, the cell media were centrifuged at 1400 rpm for 10 min at 4 °C to remove the particles. Samples were incubated with specific capture antibodies attached to magnetic beads. Subsequently, a detection antibody was introduced, followed by streptavidin-phycoerythrin, used to report the presence of the target molecule. The assay was performed in a 96-well plate using Luminex 200 instrument, and data were assessed with xPONENT 3.1. software.

### 4.7. Statistical Analysis

The statistical analysis was performed using one-way ANOVA with Tukey’s and Fisher’s LSD or non-parametric Mann–Whitney U tests. The analysis between two groups was performed by the Student’s *t*-test. Differences were considered statistically significant when *p* < 0.05. The data and statistical analysis were processed using GraphPad Prism 9.0.0 software. Each experiment was performed at least in biological triplicates.

## 5. Conclusions

The energetic metabolism of human lung microvascular endothelial cells is exclusively sensitive to SCoV2-RBD signalling. In these cells, SCoV2-RBD suppresses mitochondrial and glycolytic activity, fragments the mitochondrial network, induces mitochondrial swelling, and reduces cristae compactness, leading to cristae loss. The effects are not detectable in coronary artery endothelial or bronchial epithelial cells under the same treatment conditions.

Mitochondrial ROS mediate SCoV2-RBD-induced inflammation in lung microvascular endothelial cells, coronary artery endothelial, and bronchial epithelial cells via the GM-CSF pathway. Additionally, mitochondrial ROS are responsible for mitochondrial fragmentation and induction of IL-8 in human lung microvascular endothelial cells, IL-1β and IL-8 in coronary artery endothelial cells, and IL-1β in bronchial epithelial cells.

## Figures and Tables

**Figure 1 ijms-25-03188-f001:**
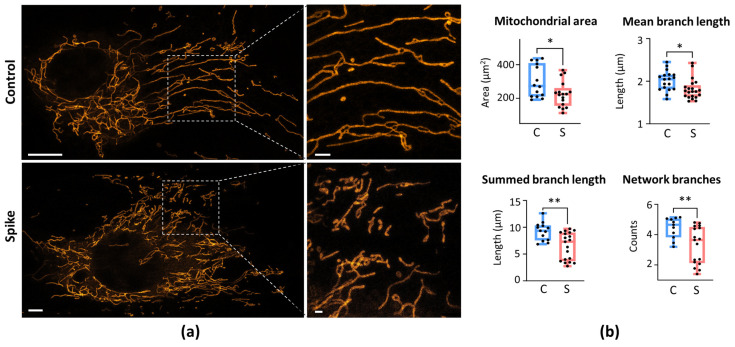
The effect of SCoV2-RBD on the mitochondrial network morphology of HLMVECs. (**a**) Representative images of the mitochondrial network of untreated (Control) and SCoV2-RBD-treated HLMVECs (Spike). Live Mito Orange and images stained the cells were taken by an Olympus IX83 confocal microscope equipped with a STEDYCON STED nanoscope. The scale bar was 5 μm for the mitochondrial network of entire cells and 1 μm for the ROIs. (**b**) The quantitative results of the mitochondrial network parameters calculated by the MiNa plugin in the Fiji software. Results were normalised to untreated cells (Control) and represented as fold changes. C—Control, S—Spike, which means SCoV2-RBD treatment. * *p* < 0.05, ** *p* < 0.01; two-tailed Student’s *t*-test.

**Figure 2 ijms-25-03188-f002:**
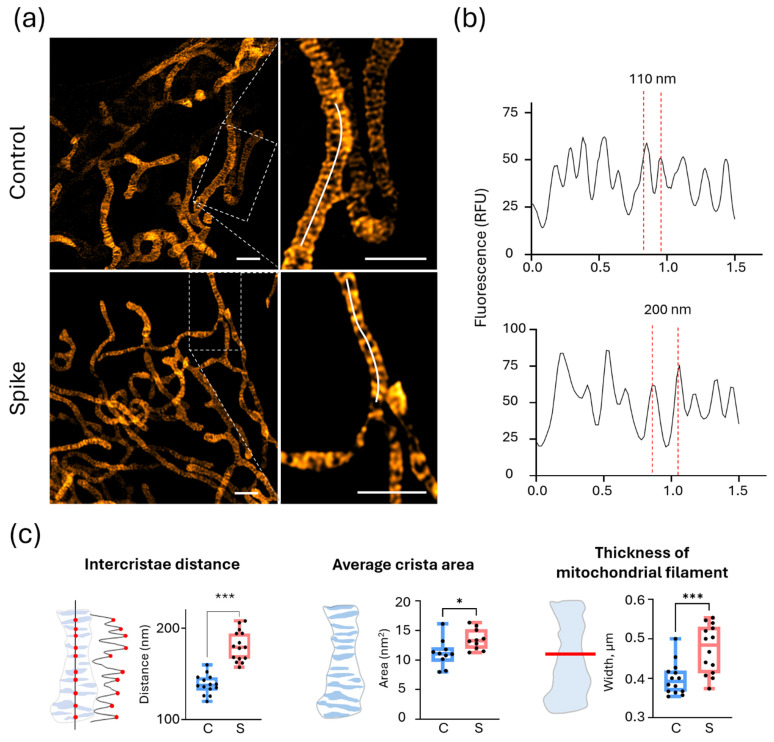
The effect of 24-h treatment of SCoV2-RBD on HLMVEC cristae density. (**a**) Representative images of untreated (Control) and SCoV2-RBD-treated (Spike) cells. The dotted line in the images on the left indicates regions of interest (ROIs) used for the fluorescence intensity profiles presented on the right. The ROIs were selected based on cristae clarity, where mitochondrial filaments were not twisted, overlapped, or out of the focal plane. The scale bar was 1 μm. (**b**) The distance between mitochondrial cristae evaluated according to the fluorescence intensity profiles measured along the curved white line on the mitochondrial network branch in (**a**), right. (**c**) Schematic representations of the calculated parameters are depicted on the left, while quantitative evaluations of intercristae distance, average crista area, and mitochondrial filament thickness are presented on the right for both the control (C) and SCoV2-RBD-treated (S) cells. Three selected fluorescence intensity profiles of 1–2 μm per cell were measured for intercristae distance evaluation in 14–16 cells per experimental group in 3 experimental repeats. For the average crista area measurements, the Trainable Weka segmentation plugin was used in Fiji. * *p* < 0.05, *** *p* < 0.001; two-tailed Student’s *t*-test.

**Figure 3 ijms-25-03188-f003:**
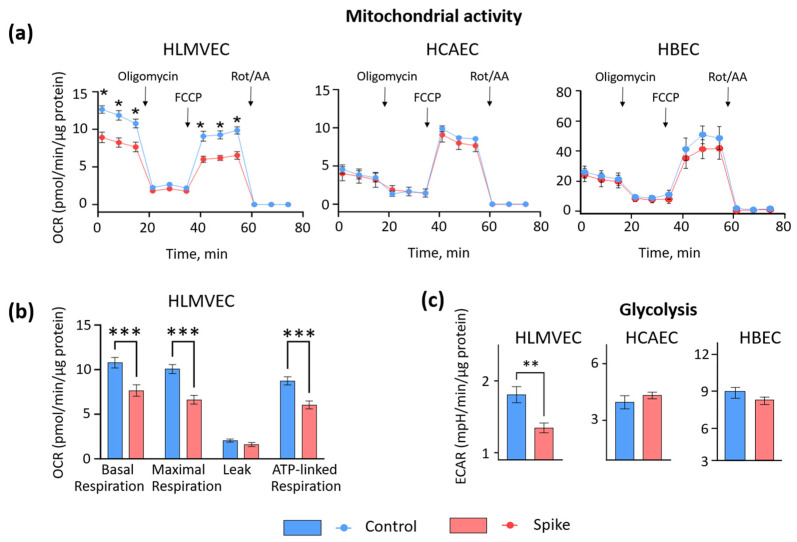
The effect of 24-h treatment of SCoV2-RBD on the mitochondrial (**a**,**b**) and glycolytic (**c**) activity of HLMVECs, HCAECs, and HBECs. (**a**) Representative mitochondrial respiration curves, OCR—oxygen consumption rate. The initial three measurements indicate the basal level of mitochondrial respiration. The next three measurements, taken after introducing oligomycin to inhibit ATP synthase, show the oxygen consumption caused by a proton leak. After that, another three measurements represent the maximum capacity of mitochondrial oxygen consumption, which occurs when FCCP uncouples the inner mitochondrial membrane. Finally, the last three measurements account for oxygen consumption unrelated to mitochondria, arising when the mitochondrial respiratory chain is hampered by rotenone and antimycin A. (**b**) Summary of the mitochondrial efficiency parameters for HLMVECs from 3 biological repeats. (**c**) Extracellular medium acidification rate (ECAR) represents the efficacy of glycolysis. * *p* < 0.05, ** *p* < 0.01, *** *p* < 0.01; one way ANOVA Tukey’s test.

**Figure 4 ijms-25-03188-f004:**
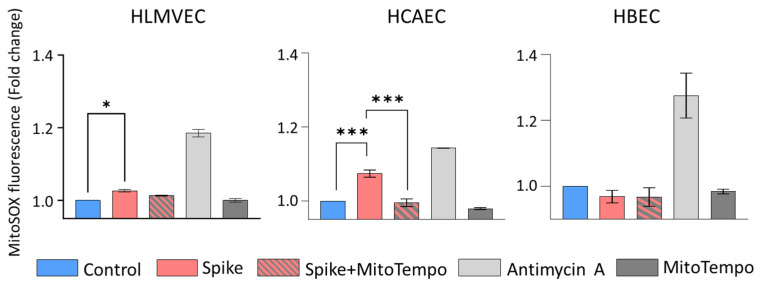
mitoROS production by HLMVECs, HCAECs, and HBECs after SCoV2-RBD treatment. Spike—SCoV2-RBD, MitoTempo—specific mitochondrial superoxide scavenger used to test assay specificity. Antimycin A—a positive control. Results were normalised to the untreated cells (Control) and represented as fold changes. * *p* < 0.05, *** *p* < 0.001; one way ANOVA Tukey’s test.

**Figure 5 ijms-25-03188-f005:**
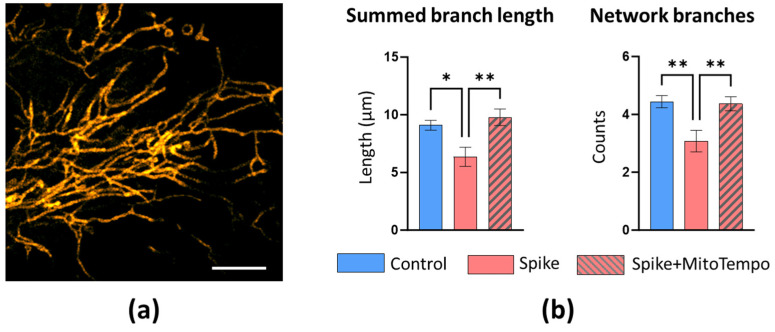
The impact of mitoROS in SCoV2-RBD-induced mitochondrial network alteration in HMLVECs. Spike—SCoV2-RBD. (**a**) Representative image of the restored HLMVEC mitochondrial network after MitoTempo addition. (**b**) Column charts represent the MitoTempo effects on quantitative mitochondrial network parameters. Results were normalised to the untreated cells (Control) and represented as fold changes. * *p* < 0.05, ** *p* < 0.01; one way ANOVA Tukey’s test.

**Figure 6 ijms-25-03188-f006:**
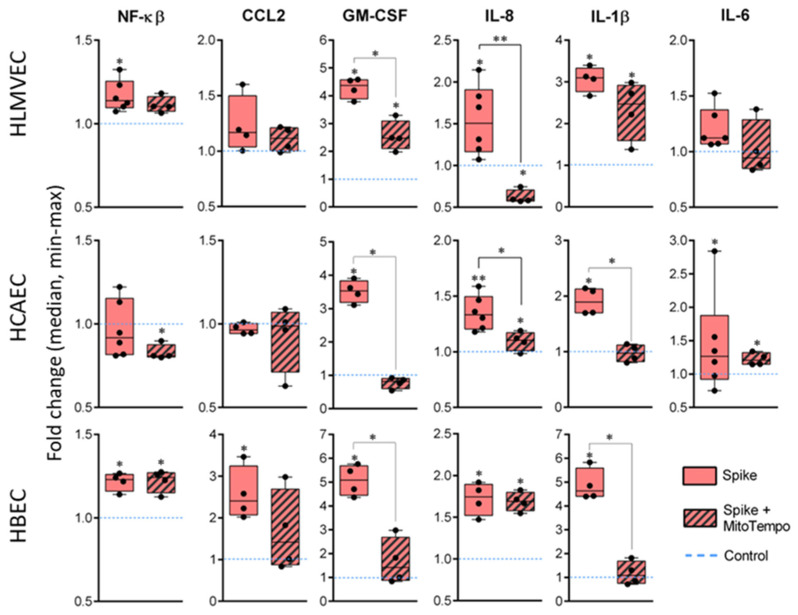
NF-kβ, CCL2, GM-CSF, IL-8, IL-1β, and IL-6 gene expression changes in HLMVECs, HCAECs, and HBECs after treatment with SCoV2-RBD (Spike). Treatment resulted in significant cytokine-coding gene expression upregulation across all cell lines. MitoTempo (Spike + MitoTempo) reversed gene expression changes caused by SCoV2-RBD in GM-CSF, IL-8, and IL-1β. Results were normalised to the untreated cells (Control) and represented as fold change values. * *p* < 0.05; ** *p* < 0.01; Mann–Whitney U test.

**Figure 7 ijms-25-03188-f007:**
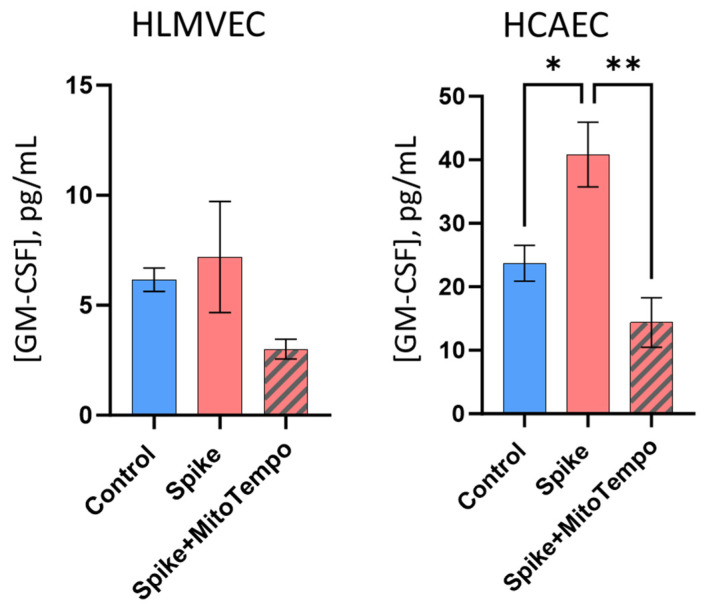
GM-CSF proinflammatory cytokine secretion levels in the HLMVEC and HCAEC cell culture medium after SCoV2-RBD alone (Spike) treatment and supplemented with MitoTempo (Spike + MitoTempo). * *p* < 0.05; ** *p* < 0.01; one way ANOVA Fisher LSD test.

## Data Availability

Data are available upon reasonable request to the corresponding author.

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
