# Peer review of "Differential Mitochondrial, Oxidative Stress and Inflammatory Responses to SARS-CoV-2 Spike Protein Receptor Binding Domain in Human Lung Microvascular, Coronary Artery Endothelial and Bronchial Epithelial Cells"

_ijms, 2024, doi:10.3390/ijms25063188_

Round 1

Reviewer 1 Report

Comments and Suggestions for Authors

In this manuscript titled “Comparison of mitochondrial response to SARS-CoV-2 spike protein receptor binding domain in human lung microvascular coronary artery endothelial and bronchial epithelial cells”, Gabrielė Kulkovienė et al. reported the effect of SARS-CoV-2 spike protein receptor binding domain on mitochondrial morphorlogy, mtROS, mitochondrial oxygen consumption, and the expression of inflammatory cytokines in human lung microvascular coronary artery endothelial and bronchial epithelial cells. The findings shown in this manuscript are potentially interesting to the readers. However, the data in the manuscript are preliminary and do not explore the relevant mechanisms.

Major comments:

1. The findings of manuscript are too preliminary, almost all the data are phenotypic and no underlying mechanisms were explored or discussed.

2. Mitochondrial structure and the intercristae distance should be analyzed by transmission electron microscopy (TEM).

3. The section of “Introduction” is short, and is needed to be supplemented.

Comments on the Quality of English Language

 Minor editing of English language required.

Author Response

Response to the comments of the Reviewer 1

Reviewer 1 – general comment:

Mitochondrial response to SARS-CoV-2 spike protein receptor binding domain in human lung microvascular coronary artery endothelial and bronchial epithelial cells, Gabrielė Kulkovienė et al. reported the effect of SARS-CoV-2 spike protein receptor binding domain on mitochondrial morphology, mtROS, mitochondrial oxygen consumption, and the expression of inflammatory cytokines in human lung microvascular coronary artery endothelial and bronchial epithelial cells. The findings shown in this manuscript are potentially interesting to the readers. However, the data in the manuscript are preliminary and do not explore the relevant mechanisms.

Authors` response: We thank the Reviewer for the feedback and appreciate the insightful comments and valuable suggestions for improvement of the clarity of the manuscript. We have considered the Reviewer's feedback and have made the necessary revisions to address recommendations, as detailed below.

Major comments:

  1. The findings of manuscript are too preliminary, almost all the data are phenotypic and no underlying mechanisms were explored or discussed.

Authors` response: We agree that exploring the mechanism of action is an essential aspect to investigate, but it would require a separate study with a different goal. In this study, we aimed to identify and compare the effects of mitochondria and its ROS participation in a SARS-CoV-2 spike glycoprotein RBD-induced inflammation in human lung microvascular, coronary artery endothelial and bronchial epithelial cells, the main cells participating in infection pathogenesis. We anticipate that our findings will stimulate investigations into molecular mechanisms that underlie the identified phenotypic changes. Therefore, our study identifies the landmarks towards further advancing the identification of potential targets for therapeutic development in the context of SARS-CoV-2 infection. 

  1. Mitochondrial structure and the intercristae distance should be analyzed by transmission electron microscopy (TEM).

Authors` response: We appreciate the constructive comment from the Reviewer regarding the additional experimental evidence of mitochondrial structure analysis by TEM. We agree that additional validation would be beneficial; however, we would like to note that the results of both methods cannot be directly comparable. Mitochondrial structure analysis by TEM occurs after fixation, and if conventional, not cryo-TEM is applied, the samples undergo intensive shrinking due to dehydration. In this study, we used superresolution STED microscopy in living cells commercially validated by the provider Abberior. Moreover, such live-cell mitochondria-staining protocols were previously validated by other scientists (https://doi.org/10.1038/s41598-019-48838-2, https://doi.org/10.1073/pnas.2215799119).

  1. The section of “Introduction” is short, and is needed to be supplemented.

Authors` response: Thank you for the suggestion. We expanded the introduction section in a revised manuscript.

Reviewer 2 Report

Comments and Suggestions for Authors

Dear authors I have had the opportunity to review your manuscript titled "Comparison of Mitochondrial Response to SARS-CoV-2 Spike Protein Receptor Binding Domain in Human Lung Microvascular, Coronary Artery Endothelial, and Bronchial Epithelial Cells" submitted to the International Journal of Molecular Sciences. Overall, your work presents valuable insights; however, I have identified a few areas that require clarification and improvement.

Introduction

·         It would be beneficial to articulate the rationale behind the selection of the three cell types more explicitly. This information will enhance the reader's understanding of the subsequent procedures.

Results

·         In Figure 1, line 64, and Figure 2, line 74, it is essential to ensure the correct statistical test is applied (t-Student or U Mann-Whitney test).

·         Based on the figure 3, it is clear that the ATP-linked respiration is also downregulated (after Oligomycin), suggesting a reduction in the ATP generation. This finding is not described in the results section, and therefore, is also not discussed.

·         Figure 3 A. Please, include a bar graph in Figure 3A quantifying basal, ATP-linked, maximal, and proton leak respiration for HLMVEC.

·         Address the temporal inconsistency where Mito Tempo is introduced in Figure 4 but is described later in figure 5. Rectify it please.

·         Mito Tempo appears for the first in the figure 4. However, the explanation of this conditions appear in figure 5.  Temporarily talking this is not correct.

·         Line 118-119: “However, mitoROS had no detectable impact on SCoV2-RBD effect on mitochondrial footprint area and mean branch length (Fig. A5), indicating that mitoROS were not in control of SCoV2-RBD-induced mitochondrial loss.” This phrase is confused and needs clarification.

·         Line 153. “Despite cytokine expression-attenuation effects” this phrase is unclear. It should include Mito-tempo somewhere. I suggest: “Despite the attenuating effects of mitoTempo on cytokines expression, GM-CSF…”

·         Figure 6 shows that NF-Kb and IL-8 remained significant after Mito-tempo treatment in HBEC. However in line 155 you declare that IL-1B also remained significant. Please clarify.

Discussion

·         Lines 171 – 172: “suppressed basal and uncoupler-stimulated mitochondrial respiration, pointing to inhibition of the mitochondrial respiratory chain”.  This conclusion is a bit exaggerated. The evidence point to that there is a reduction in the oxygen consumption, however, the cause is not investigated here. A reduction in OCR could also be related to reduced substrate availability, altered function of Krebs cycle, ATP syntheses inhibition, among others. Please, moderate the conclusion or suggest other potential causes for reduced OCR.

·         Lines 178 – 196: This paragraph is discussing something that is not proved here. There is no direct evidence of ETC inhibition, changes in OXPHOS expression, or in super-complexes formation that explains the cause of OCR reduction. Please, focus the discussion on the direct evidence from your data and avoid speculating about unmeasured factors.

·         Line 231 – 234 “ACE2 binding can trigger mitoROS via NOX4 [17]; thus, the descending mitoROS induction in the row of HCAEC, HLMVEC, and HBEC can at least partly be explained by higher expression of ACE2 in cardiovascular tissues compared to the pulmonary and in endothelium compared to the epithelium [18,19].” Reevaluate and simplify it. Your set up is different to other studies and did not include a control in cardiac cell lines.

·         Line 254: the word “function” by “functional”.

Others:

·         Please, define the acronym GM-CSF for non-experts in cytokines.

I appreciate your efforts in contributing to the field, and I believe that addressing these points will significantly improve the clarity and impact of your manuscript. I look forward to seeing the revised version.

Comments on the Quality of English Language

the English language usage in the article may benefit from minor editing for clarity, and a few improvements in phrasing could enhance overall readability. Furthermore, it is suggested that the Discussion section be revised to enhance the fluency and readability of the content. Consider incorporating subheadings to improve the overall structure and facilitate a clearer presentation of ideas.

Author Response

Response to the comments of the Reviewer 2

Reviewer 2 – general comment:

Dear authors I have had the opportunity to review your manuscript titled "Comparison of Mitochondrial Response to SARS-CoV-2 Spike Protein Receptor Binding Domain in Human Lung Microvascular, Coronary Artery Endothelial, and Bronchial Epithelial Cells" submitted to the International Journal of Molecular Sciences. Overall, your work presents valuable insights; however, I have identified a few areas that require clarification and improvement.

Authors' response: We thank the Reviewer for the thorough and thoughtful review of our manuscript. We appreciate the insightful comments and valuable suggestions for improvement of the clarity of the manuscript. We have considered the Reviewer's feedback and have made the necessary revisions to address recommendations, as detailed below.

Introduction

It would be beneficial to articulate the rationale behind the selection of the three cell types more explicitly. This information will enhance the reader's understanding of the subsequent procedures.

Authors' response: We appreciate your suggestion and have supplemented the Introduction section providing justification for cell type selection. Shortly, we selected the cells according to their involvement in COVID-19 pathophysiology. The first target of coronaviruses is the airway; however, cardiovascular tissues can also be involved in disease pathogenesis. Moreover, all types of selected cells have expressed ACE2 receptors necessary for SARS-CoV-2 entry into the cell, supposing that the virus can directly affect these cell types. In our study, we compared the effects caused by SARS-CoV-2 spike RBD between the two airway-blood barrier-forming and cardiovascular cell types.

Results

In Figure 1, line 64, and Figure 2, line 74, it is essential to ensure the correct statistical test is applied (t-Student or U Mann-Whitney test).

Authors' response: We appreciate your observation and have corrected both figures' descriptions. The correct statistical test was the Student's t-test.

Based on the figure 3, it is clear that the ATP-linked respiration is also downregulated (after Oligomycin), suggesting a reduction in the ATP generation. This finding is not described in the results section, and therefore, is also not discussed.

Authors' response: Thank you for the observation. Indeed, ATP-linked respiration in HLMVEC was significantly reduced after treatment with SARS-CoV-2 spike protein receptor binding domain. Accordingly, the results (line 99) and discussion (line 207) parts were supplemented.

Figure 3 A. Please, include a bar graph in Figure 3A quantifying basal, ATP-linked, maximal, and proton leak respiration for HLMVEC.

Authors' response: Bar graphs of basal, maximal, ATP-linked respiration and leak were added to Fig.3 graph as section b. 

Address the temporal inconsistency where Mito Tempo is introduced in Figure 4 but is described later in figure 5. Rectify it please.

Authors' response: The introduction of MitoTempo was moved to line 121, before Figure 4, where it appears first.

Mito Tempo appears for the first in the figure 4. However, the explanation of this conditions appear in figure 5.  Temporarily talking this is not correct.

Authors' response: The explanation of MitoTempo conditions was moved to the section before Fig. 4. 

Line 118-119: "However, mitoROS had no detectable impact on SCoV2-RBD effect on mitochondrial footprint area and mean branch length (Fig. A5), indicating that mitoROS were not in control of SCoV2-RBD-induced mitochondrial loss." This phrase is confused and needs clarification.

Authors' response: We have rephrased this part of the results, adding more explanation about the different mitochondrial parameters measured (Lines 148-156). 

Line 153. "Despite cytokine expression-attenuation effects" this phrase is unclear. It should include Mito-tempo somewhere. I suggest: "Despite the attenuating effects of mitoTempo on cytokines expression, GM-CSF…"

Authors' response: The phrase was corrected according to the recommendation (Line 187). 

Figure 6 shows that NF-Kb and IL-8 remained significant after Mito-tempo treatment in HBEC. However in line 155 you declare that IL-1B also remained significant. Please clarify.

Authors' response: Thank you for the observation. In HBEC cells, only NF-Kb and IL-8 remained significantly upregulated. We corrected a mistake (Line 189). 

Discussion 

Lines 171 – 172: "suppressed basal and uncoupler-stimulated mitochondrial respiration, pointing to inhibition of the mitochondrial respiratory chain".  This conclusion is a bit exaggerated. The evidence point to that there is a reduction in the oxygen consumption, however, the cause is not investigated here. A reduction in OCR could also be related to reduced substrate availability, altered function of Krebs cycle, ATP syntheses inhibition, among others. Please, moderate the conclusion or suggest other potential causes for reduced OCR.

Authors' response: Thank you for the correct insight. We have moderated the conclusion following the suggestion. However, we are not so sure about the possibility of ATP synthesis inhibition. If it were the cause of the SCoV2-RBD-mediated mitochondrial respiration suppression in HLMVEC, it would be visible as a difference between the effect on coupled and uncoupled respiration rates. 

Lines 178 – 196: This paragraph is discussing something that is not proved here. There is no direct evidence of ETC inhibition, changes in OXPHOS expression, or in super-complexes formation that explains the cause of OCR reduction. Please, focus the discussion on the direct evidence from your data and avoid speculating about unmeasured factors.

Authors' response: We have removed the paragraph about supercomplexes from the Discussion.

Line 231 – 234 "ACE2 binding can trigger mitoROS via NOX4 [17]; thus, the descending mitoROS induction in the row of HCAEC, HLMVEC, and HBEC can at least partly be explained by higher expression of ACE2 in cardiovascular tissues compared to the pulmonary and in endothelium compared to the epithelium [18,19]." Reevaluate and simplify it. Your set up is different to other studies and did not include a control in cardiac cell lines.

Authors' response: We considered your suggestion and paraphrased (Lines 259-262). We would like to draw attention to the fact that we have used primary human coronary artery cells in our study.

Line 254: the word "function" by "functional".

Authors' response: The word was corrected according to the observation. 

Others:

Please, define the acronym GM-CSF for non-experts in cytokines.

Authors' response: The acronym is now defined in the abstract of a manuscript. 

I appreciate your efforts in contributing to the field, and I believe that addressing these points will significantly improve the clarity and impact of your manuscript. I look forward to seeing the revised version.

Comments on the Quality of English Language

the English language usage in the article may benefit from minor editing for clarity, and a few improvements in phrasing could enhance overall readability. Furthermore, it is suggested that the Discussion section be revised to enhance the fluency and readability of the content. Consider incorporating subheadings to improve the overall structure and facilitate a clearer presentation of ideas.

Authors' response: Thank you for observations. We have added subheadings in the discussion section, as recommended.

Reviewer 3 Report

Comments and Suggestions for Authors

Authors provide comparison of mitochondrial response to SARS-CoV-2 spike protein receptor binding domain in human lung microvascular, coronary artery endothelial and bronchial epithelial cells. They tested the effect of SARS-CoV-2 receptor binding domain (RBD) on the mitochondrial network and cristae morphology, oxygen consumption, mitoROS production, and inflammatory cytokine expression in cultured human lung microvascular (HLMVEC) and coronary artery endothelial (HCAEC) and bronchial epithelial cells (HBEC). Mitochondrial ROS control SARS-CoV-2 RBD-induced inflammation in HLMVEC, HCAEC and HBEC, with the mitochondria of HLMVEC being more sensitive to SARS-CoV-2 RBD.

This is an interesting high quality paper. Thera are few minor suggestions to improve manuscript.

Minor

1. The paper sounds too descriptive despite many mechanistic studies. It is recommended to modify the title to reflect the mechanistic insights, for example, 

Differential responses to SARS-CoV-2 spike protein receptor binding domain in human lung microvascular, coronary artery endothelial and bronchial epithelial cells on mitochondrial function, inflammation and oxidative stress. 

2. The Introduction section is extremely short. This paper would benefit from expanded Introduction.  Authors may add a couple of paragraph a) describing an important role of mitochondria in endothelial function, and b) discussing pathophysiological role of ROS and inflammation in endothelial cells. These 2 paragraphs are not intended for citing ARS-CoV-2 research but must help with study justification (mitochondria, inflammation etc.) as well as help the readers to put the presented data in the mechanistic pathophysiological  context. 

3. Method section provides Gene expression methods used for cytokine studies in Figures 6 and 7, however, there is no method section for proinflammatory cytokine secretion levels analysis (Figure A6). Please add this paragraph to the Method section.

4. There are couple of minor issues with the experimental methods.

4A. Authors may comment in the manuscript that MitoSOX fluorescence is not fully specific for mitochondrial superoxide due to formation of superoxide specific adduct/product (mito-2-hydroxy-ethidium) and unspecific oxidation product (mito-ethidium). See PMID: 23668959.

4B. It is also interesting if authors tested that MitoSOX fluorescence was inhibited by MitoTempo. It is useful to cite the original paper for design and validation of MitoTempo in endothelial cells: Dikalova et al. Circ Res. 2010 Jul 9;107(1):106-16. 

Please note that in endothelial cell culture the optimal concentration for MitoTempo is 50 nM but not the 10 uM used by Authors since this high dose is not mitochondria specific, overload mitochondria and scavenge both mitochondria and cytoplasmic superoxide.

Author Response

Response to the comments of the Reviewer 3

Reviewer 3 – general comment:

Authors provide comparison of mitochondrial response to SARS-CoV-2 spike protein receptor binding domain in human lung microvascular, coronary artery endothelial and bronchial epithelial cells. They tested the effect of SARS-CoV-2 receptor binding domain (RBD) on the mitochondrial network and cristae morphology, oxygen consumption, mitoROS production, and inflammatory cytokine expression in cultured human lung microvascular (HLMVEC) and coronary artery endothelial (HCAEC) and bronchial epithelial cells (HBEC). Mitochondrial ROS control SARS-CoV-2 RBD-induced inflammation in HLMVEC, HCAEC and HBEC, with the mitochondria of HLMVEC being more sensitive to SARS-CoV-2 RBD.

This is an interesting high quality paper. Thera are few minor suggestions to improve manuscript.

Authors' response: We thank the Reviewer for the thorough and thoughtful review of our manuscript. We appreciate the insightful comments and valuable suggestions for improvement of the clarity of the manuscript. We have considered the Reviewer's feedback and have made the necessary revisions to address recommendations, as detailed below.

Minor

  1. The paper sounds too descriptive despite many mechanistic studies. It is recommended to modify the title to reflect the mechanistic insights, for example, 

Differential responses to SARS-CoV-2 spike protein receptor binding domain in human lung microvascular, coronary artery endothelial and bronchial epithelial cells on mitochondrial function, inflammation and oxidative stress.

Authors' response: We agree to correct the manuscript title based on your suggestion.

  1. The Introduction section is extremely short. This paper would benefit from expanded Introduction.  Authors may add a couple of paragraph a) describing an important role of mitochondria in endothelial function, and b) discussing pathophysiological role of ROS and inflammation in endothelial cells. These 2 paragraphs are not intended for citing ARS-CoV-2 research but must help with study justification (mitochondria, inflammation etc.) as well as help the readers to put the presented data in the mechanistic pathophysiological context.

Authors' response: Thank you for your suggestions. We supplemented the Introduction by adding additional information to provide context for the study about mitochondrial involvement in inflammation and the rationale of selected cell lines.

  1. Method section provides Gene expression methods used for cytokine studies in Figures 6 and 7, however, there is no method section for proinflammatory cytokine secretion levels analysis (Figure A6). Please add this paragraph to the Method section.

Authors' response: Thank you for noticing. We have added a 4.6. paragraph on cytokine secretion analysis in the Method section.  

  1. There are couple of minor issues with the experimental methods. 

4A. Authors may comment in the manuscript that MitoSOX fluorescence is not fully specific for mitochondrial superoxide due to formation of superoxide specific adduct/product (mito-2-hydroxy-ethidium) and unspecific oxidation product (mito-ethidium). See PMID: 23668959.

Authors' response: We have commented on the use of MitoSOX in the method section (4.4 Mitochondrial superoxide).  We considered the limitations of MitoSOX, and thus, we implemented control measures and used the minimum possible MitoSOX concentration, as suggested by Murphy and colleagues in their research paper (doi.org/10.1038/s42255-022-00591-z). A mitochondria-specific antioxidant, MitoTempo, was used as a negative control, while Antimycin A, which inhibits electron transport chain complex III and induces superoxide production, served as a positive control.

4B. It is also interesting if authors tested that MitoSOX fluorescence was inhibited by MitoTempo. It is useful to cite the original paper for design and validation of MitoTempo in endothelial cells: Dikalova et al. Circ Res. 2010 Jul 9;107(1):106-16. 

Authors' response: Yes, we have tested MitoTempo inhibition of MitoSOX fluorescence. MitoTempo tended to decrease MitoSOX fluorescence by 2 % compared to the control. Thank you for the suggestion to cite the original paper on the design and validation of MitoTempo, we have referenced it in the Method section. 

Please note that in endothelial cell culture the optimal concentration for MitoTempo is 50 nM but not the 10 uM used by Authors since this high dose is not mitochondria specific, overload mitochondria and scavenge both mitochondria and cytoplasmic superoxide.

Authors' response: We thank you for this insight. Certainly, the MitoTempo overload would lead unspecific reaction, which should be avoided. However, the study you are referring to (Dikalova et al. Circ Res. 2010 Jul 9;107(1):106-16) has examined MitoTempo accumulation and its optimal concentration of mitochondrial superoxide dismutation on bovine aortic endothelial cells. Different species and cell lines may exhibit variations in sensitivity to MitoTempo. For instance, concentration for endothelial cells from the heart of mice was used at 10 µM, as reported in a 2021 paper by Song Yi et al ( doi.org/10.1016/j.jtcvs.2021.06.029). Cardiac myocytes derived from human iPSC used a concentration of 25 µM (Li A et al, 2022, doi.org/10.1016/j.freeradbiomed.2022.08.005). 10 µM was also used for epithelial cells from rat kidneys (Zhang J et al, 2017,  https://doi.org/10.1155/2017/7528090) and from human colon tumors (Wang A et al, 2014, doi.org/10.1016/j.ajpath.2014.05.019). Based on these latest studies, we screened MitoTempo efficiency on our cells subjected to Antimycin A and chose 10 µM as the optimal concentration for endothelial and epithelial cell lines. 

Round 2

Reviewer 1 Report

Comments and Suggestions for Authors

The revised manuscript quality has not been significant improved, and most of my concerns were not addressed and resolved. The findings of manuscript are still too preliminary, almost all the data are phenotypic and no underlying mechanisms were further explored.

Author Response

We agree the study is phenotypic, and molecular mechanisms were not intensively investigated. We aimed to compare mitochondrial responses to SARS-CoV-2-RBD signalling in different human cell types most affected during SARS-CoV-2 infection, specifically focusing on mitochondrial morphometrics in living cells down to the cristae level. The study found that primary lung microvascular endothelial cells undergo mitochondrial function, network and cristae remodelling after SARS-CoV-2 spike glycoprotein RBD stimulation and did confirm the role of mitochondrial ROS in SARS-CoV-2-RBD-induced mitochondrial fragmentation and inflammatory cytokine secretion. We believe the findings serve as a background for future research in determining the role of mitochondrial network and cristae dynamics in viral inflammatory signalling.